# Bioactive Dimeric Abietanoid Peroxides from the Bark of *Cryptomeria japonica*

**DOI:** 10.3390/molecules24112178

**Published:** 2019-06-10

**Authors:** Chi-I Chang, Cheng-Chi Chen, Chiy-Rong Chen, Ming-Der Wu, Ming-Jen Cheng, Ping-Jyun Sung, Yueh-Hsiung Kuo

**Affiliations:** 1Department of Biological Science and Technology, National Pingtung University of Science and Technology, Pingtung 912, Taiwan; changchii@mail.npust.edu.tw; 2Research Center for Active Natural Products Development, National Pingtung University of Science and Technology, Pingtung 912, Taiwan; 3Department of Chemistry, National Taiwan University, Taipei 106, Taiwan; r93223083@ntu.edu.tw; 4Department of Life Science, National Taitung University, Taitung 950, Taiwan; gina77@nttu.edu.tw; 5Bioresource Collection and Research Center (BCRC), Food Industry Research and Development Institute (FIRDI), Hsinchu 300, Taiwan; wmd@firdi.org.tw (M.-D.W.); cmj@firdi.org.tw (M.-J.C.); 6National Museum of Marine Biology and Aquarium, Pingtung 944, Taiwan; pjsung@nmmba.gov.tw; 7Graduate Institute of Marine Biology, National Dong Hwa University, Pingtung 944, Taiwan; 8Department of Chinese Pharmaceutical Sciences and Chinese Medicine Resources, College of Pharmacy, China Medical University, Taichung 404, Taiwan; 9Department of Biotechnology, Asia University, Taichung 413, Taiwan; 10Chinese Medicine Research Center, China Medical University, Taichung 404, Taiwan

**Keywords:** Cupressaceae, *Cryptomeria japonica*, dimeric abietane, diterpenoid

## Abstract

Three new dimeric abietane-type diterpenoids, abieta-6,8,11,13-tetraen-12-yl 12-hydroxyabieta-8,11,13-trien-7α-yl peroxide (**1**), abieta-6,8,11,13-tetraen-12-yl 12-hydroxyabieta-8,11,13-trien-7β-yl peroxide (**2**), and 12-hydroxyabieta-8,11,13-trien-7β-yl 7-oxoabieta-5,8,11,13-tetraen-12-yl peroxide (**3**), together with four known abietane-type diterpenoids (**4**–**7**) were isolated from the methanol extract of the bark of *Cryptomeria japonica*. Their structures were elucidated on the basis of spectroscopic analysis and comparison of NMR data with those of known analogues. At a concentration of 50 μM, compounds **1**, **2**, and **3** showed 26.2%, 23.6%, and 35.7% inhibition towards xanthine oxidase enzyme, respectively. In addition, compound **3** also showed 24.9% inhibition toward angiotensin-converting enzyme (ACE).

## 1. Introduction

The Japanese cedar, *Cryptomeria japonica* D. Don (Cupressaceae), belongs to the monotypic genus in the Cupressaceae [1]. It is a massive evergreen coniferous tree, growing up to 50 m in height. Due to the aromatic, reddish-pink in color, soft, lightweight but strong, and waterproof properties of its wood, it is planted in large quantities and used as building materials and wood products. This plant has been cultivated as an important coniferous tree species in Taiwan since 1906. Phytochemical investigations indicated the presence of monoterpenoids, sesquiterpenoids, and diterpenoids [2,3,4,5,6,7,8,9,10,11,12,13,14,15,16,17,18,19,20,21,22,23,24] in the leaves, heartwood, and barks of *C. japonica*. The crude extracts and secondary metabolites from this species exhibited a wide array of bioactivities including cytotoxic [23], antifungal [24], antibacterial [23], antioxidant [25], anti-inflammatory [26], and insect antifeedant [27] and repellent [18] properties. While searching for bioactive chemical ingredients of the bark of *C. japonica*, we have already reported the isolation of three sesquarterpenoids [28,29] and ten abietane-type diterpenoids [30,31,32]. Herein, the isolation and structure elucidation of three new dimeric abietane-type diterpenoids are described as well as their inhibitory activity towards xanthine oxidase and angiotensin-converting enzymes.

## 2. Results and Discussion

### 2.1. Isolation and Structural Elucidation

The MeOH extract of the bark of *C. japonica* was suspended in H_2_O and then partitioned with EtOAc and *n*-BuOH, successively. The EtOAc-soluble portion was submitted to repeated silica gel column chromatography and semipreparative normal phase-HPLC to afford compounds **1**–**7** (Figure 1).

The high resolution electron impact mass spectrum (HR-EI-MS) of **1** gave a molecular ion at *m*/*z* 584.4238, corresponding to the molecular formula of C_40_H_56_O_3_, with thirteen degrees of unsaturation. The electron impact mass spectrum (EI-MS) of **1** displayed the fragmental ions at *m*/*z* 300 [C_20_H_28_O_2_]^+^ and 284 [C_20_H_28_O]^+^ (Figure 2) and forty carbon signals were observed in the ^13^C-NMR spectrum, indicating that **1** was a dimeric diterpenoid (Figure 2). The UV and IR spectra showed absorption bands for hydroxyl (3409 cm^−1^) and aromatic (λ_max_ 220 and 279 nm; 3049, 1593, and 1487 cm^−1^) groups. The ^1^H- and ^13^C-NMR data of **1** (Table 1) showed one set of dehydroabietane diterpene signals for constituent upper monomer-1 including three tertiary methyl groups (δ_H_ 0.97, 0.98, and 1.36 (each 3H, s, Me-18, Me-19, and Me-20)), an isopropyl group attached to a phenyl group (δ_H_ 0.78 (3H, d, *J* = 7.0 Hz, Me-16), 1.01 (3H, d, *J* = 7.0 Hz, Me-17), and 2.85 (1H, sept, *J* = 7.0 Hz, H-15)), two para aromatic protons (δ_H_ 6.34 (1H, s, H-14) and 6.69 (1H, s, H-11)), a phenolic hydroxyl proton (δ_H_ 4.33 (1H, s, exchangeable with D_2_O)), and a typical downshifted H_β_-1 signal of a dehydroabietane diterpene (δ_H_ 1.96 (1H, br d, *J* = 12.0 Hz)) [33]. A downshifted benzyl proton connected with a peroxyl group (δ_H_ 5.86 (1H, br s, H-7)), instead of a hydroxyl group [19,33,34], was assigned as H-7, suggested by the ^1^H-^1^H COSY correlations with the two methylene protons of H-6 (δ_H_ 2.28 (1H, m), δ_H_ 2.32 (1H, m)), and HMBC correlations with C-5 (δ_C_ 43.5) and C-8 (δ_C_ 145.6; Figure 3). In addition, the ^1^H-NMR signal of H-7 was a broad singlet peak and showed the NOESY correlation with both H_α_-6 (δ_H_ 2.32) and H_β_-6 (δ_H_ 2.28), hinted that the peroxyl group was attached on C-7 in α-axial orientation [33] (Figure 3). These data proved that the structure of constituent monomer-1 was related to 7α-peroxyferruginol. The ^1^H- and ^13^C-NMR data of **1** (Table 1) also exhibited another set of dehydroabietane diterpene signals for constituent lower monomer-2 including three tertiary methyl groups (δ_H_ 0.94, 1.04, and 1.11 (each 3H, s, Me-18′, Me-19′, and Me-20′)), an isopropyl group attached to a phenyl group (δ_H_ 0.81 (3H, d, *J* = 7.0 Hz, Me-16′), 0.98 (3H, d, *J* = 7.0 Hz, Me-17′), and 2.82 (1H, sept, *J* = 7.0 Hz, H-15′)), two para aromatic protons (δ_H_ 6.98 (1H, s, H-11′) and 6.77 (1H, s, H-14′)), an ABX coupling system of one methine proton (δ_H_ 2.07 (1H, dd, *J* = 3.0, 2.5 Hz, H-5′)), and two vinyl protons (δ_H_ 5.87 (1H, dd, *J* = 9.5, 2.5 Hz, H-6′) and 6.45 (1H, dd, *J* = 9.5, 3.0 Hz, H-7′)), together with a typical downshifted H_β_-1 signal of a dehydroabietane diterpene (δ_H_ 2.22 (1H, br d, *J* = 13.0 Hz)) [33]. The above NMR spectroscopic data suggested constituent lower monomer-2 was related to 6,7-dehydroferruginol [35]. Thus, the gross structure of **1** is composed of 7α-hydrperoxyferruginol and 6,7-dehydroferruginol. The chemical shift of H-7 appeared at the lower field region (δ_H_ 5.86) in constituent monomer-1, comparing to that of the 7α-hydroxyferruginol analogues [34] implied that the connectivity of two monomers between C-7 and C-12′ through a peroxide functionality. The NOESY correlation between H-7 and H-11′ (δ_H_ 6.98; Figure 3) further confirmed this proposal. Interestingly, there are some electrostatic attraction between the electron-rich aryl, the phenol functionality of upper 7α-peroxyferruginol derivative, and the electron-deficient aryl, with peroxide moiety of lower 6,7-dehydroferruginol, formed the most stable conformer as shown in Figure 3. Due to the anisotropic effect from the opposite phenyl group, H-14, H-15, H-16, H-17, H-14′, H-15′, H-16′, H-17′, and phenol of **1** were posited in the shielding region and thus showed the higher field chemical shifts than that of the usual dehydroabietane diterpene. In contrast, H-5, H-7, and H-20’ of **1** were located in the deshielding region and thus exhibit a lower field chemical shifts than that of the usual dehydroabietane diterpene [33]. Complete ^1^H- and ^13^C-NMR chemical shifts were established by ^1^H-^1^H COSY, HMQC, HMBC, and NOESY spectra. Based on these above evidences, compound **1** was elucidated as abieta-6,8,11,13-tetraen-12-yl 12-hydroxyabieta-8,11,13-trien-7α-yl peroxide. 

Compound **2** was an isomer of **1** with the same molecular formula C_40_H_56_O_3_, determined by the molecular ion of HR-EI-MS at *m*/*z* 584.4237. Its EI-MS also showed the fragmental ions at *m*/*z* 300 [C_20_H_28_O_2_]^+^ and 284 [C_20_H_28_O]^+^, indicating that 2 was also a dimeric diterpenoid. The absorptions for hydroxyl (3423 cm^−1^) and aromatic (3048, 1590, and 1493 cm^−1^; λ_max_ 217 and 276 nm) groups were also found in the UV and IR spectra. Comparison of ^1^H and ^13^C-NMR data of 2 and **1** (Table 1) showed that the signals of constituent lower monomer-2 of **2** were almost the same as those of **1**, indicating the structure of constituent lower monomer-2 is related to 6,7-dehydroferruginol. The ^1^H- and ^13^C-NMR data of **2** (Table 1) also showed another set of dehydroabietane diterpene signals for constituent upper monomer-1 including three tertiary methyl groups (δ_H_ 0.95, 1.02, and 1.37 (each 3H, s, Me-18, Me-19, and Me-20)), an isopropyl group attached to a phenyl group (δ_H_ 1.16 (3H, d, *J* = 7.0 Hz, Me-17), 1.18 (3H, d, *J* = 7.0 Hz, Me-16), and 3.01 (1H, sept, *J* = 7.0 Hz, H-15)), two para aromatic protons (δ_H_ 6.72 (1H, s, H-14) and 6.74 (1H, s, H-11)), a phenolic hydroxyl proton (δ_H_ 4.52 (1H, s)), and a typical downshifted H_β_-1 signal of a dehydroabietane diterpene (δ_H_ 1.93 (1H, br d, *J* = 12.0 Hz)) [33]. A downshifted benzyl proton connected with a peroxyl group (δ_H_ 5.20 (1H, d, *J* = 8.0 Hz, H-7)) was assigned as H-7, suggesting by the ^1^H-^1^H COSY correlations with the two methylene protons of H-6 (δ_H_ 2.24 (1H, m, H_α_-6), δ_H_ 2.40 (1H, br d, *J* = 8.0 Hz, H_β_-6)) and HMBC correlations with C-5 (δ_C_ 47.6) and C-8 (δ_C_ 147.2; Figure 3). Since the ^1^H-NMR signal of H-7 was a doublet peak with a constant coupling constant, 8.0 Hz, the peroxyl group on C-7 was in β-equational orientation (Figure 2), instead of in α-axial orientation in **1** [33]. H-7 showed the NOESY correlation with H_α_-6 (δ_H_ 2.24) and 1.59 (H-5, m), but the lack of NOESY correlation with H_β_-6 (δ_H_ 2.40) further confirmed this proposal (Figure 3). Thus, the structure of 2 was identified as abieta-6,8,11,13-tetraen-12-yl 12-hydroxyabieta-8,11,13-trien-7β-yl peroxide. Compound **2** did not exhibit the most stable conformer as in **1**. Therefore, the chemical shifts of two isopropyl groups in **2** were not shifted to the high field region. 

The HR-EI-MS of 3 showed a molecular ion at *m*/*z* 598.4018, which corresponded to the molecular formula, C_40_H_54_O_4_, indicating fourteen degrees of unsaturation. The EI-MS fragmental ions of **3** at *m*/*z* 300 [C_20_H_28_O_2_]^+^ and 298 [C_20_H_26_O_3_]^+^ indicated that **3** was also a dimeric diterpenoid. The absorptions for hydroxyl (3376 cm^−1^), benzoyl (1639 cm^−1^; λ_max_ 233, 282, and 310 nm) and aromatic (3049, 1586, and 1467 cm^−1^) groups were also found in its UV and IR spectra. Comparison of ^1^H- and ^13^C-NMR data of 3 and **2** (Table 1) showed that the signals of constituent monomer-1 of **3** were almost the same as those of **2**, indicating the structure of constituent upper monomer-1 is related to 7α-peroxyferruginol. The ^1^H- and ^13^C-NMR data of **3** (Table 1) also showed another set of dehydroabietane diterpene signals for constituent lower monomer-2 as follows: Three tertiary methyl groups (δ_H_ 0.97, 1.03, and 1.49 (each 3H, s, Me-18′, Me-19′, and Me-20)), an isopropyl group attached to a phenyl group (δ_H_ 1.20 (3H, d, *J* = 7.0 Hz, Me-17’), 1.23 (3H, d, *J* = 7.0 Hz, Me-16’) and 3.32 (1H, sept, *J* = 7.0 Hz, H-15′)), two para aromatic protons (δ_H_ 7.19 (1H, s, H-11′) and 8.03 (1H, s, H-14′)), one trisubstituted double bond (δ_H_ 6.48 (1H, s, H-6′); δ_C_ 124.5 (C-6′), 173.2 (C-5′)), and a typical downshifted H_β_-1 signal of a dehydroabietane diterpene (δ_H_ 2.51 (1H, br d, *J* = 13.5 Hz, H-1′)). The NMR spectroscopic data of constituent lower monomer-2 showed a close structural resemblance to that of 5,6-dehydrosugiol [36]. Thus, the structure of constituent lower monomer-2 was tentatively determined as 5,6-dehydrosugiol-related abietane. H-7 exhibited a doublet of a doublet signal with two coupling constants, 9.0 and 2.5 Hz and showed the NOESY correlation with H_α_-6 (δ_H_ 2.24) and H-5 (δ_H_ 1.72), but the lack of NOESY correlation with H_β_-6 (δ_H_ 2.41), which confirmed the peroxyl group was attached on C-7 in β-equational orientation (Figure 2) [33]. Thus, the structure of **3** was identified as 12-hydroxyabieta-8,11,13-trien-7β-yl 7-oxoabieta-5,8,11,13-tetraen-12-yl peroxide. 

Four known compounds were identified by comparison of the NMR data with those described in the literatures as sugiol (**4**) [37], 16-hydroxysugiol (**5**) [38], 12-dehydroxy-15-hydroxysugiol (**6**) [39], and 12-*O*-acetylsugiol (**7**) [40].

### 2.2. Inhibitory Activities Toward Xanthine Oxidase and Angiotensin-Converting Enzyme

Xanthine oxidase is a key enzyme in purine metabolic pathway, catalyzing oxypurines (hypoxanthine and xanthine) to uric acid and plays an important role in causing gout [41]. Additionally, the angiotensin-converting enzyme (ACE) plays a key physiological role in blood pressure regulation of the renin–angiotensin system due to its action in the formation of angiotensin II, a potent vasoconstrictor, and in the degradation of bradykinin, a vasodilator [42]. Compounds **1**–**3** were evaluated using the above two enzyme inhibitory activities [43,44]. At the concentration of 50 μM, compounds **1**–**3** exhibited 26.2%, 23.6%, and 35.7% xanthine oxidase inhibitory activity, respectively. Compound **3** also showed 24.9% ACE inhibitory activity, while compounds **1** and **2** were inactive. Analysis of the relationship between structure and activity in compounds **1**–**3** showed that the inhibitory activities toward the two above enzymes of compound **3** containing a 5,6-dehydrosugiol moiety at C-7 were higher than that of compounds **1** and **2** with a 6,7-dehydroferruginol moiety at C-7. Furthermore, the different orientations of the 7-substituent between compounds **1** and **2** had no significant effect on their xanthine oxidase inhibitory activity.

## 3. Experimental Section

### 3.1. Chemicals

Xanthine, Xanthine oxidase, and ACE (EC 3.4.15.1) from rabbit lungs, hippuryl-l-histidyl-l-leucine (HHL), ferulic acid (FA), sodium chloride (NaCl), and sodium hydroxide (NaOH) were purchased from Sigma Chemical Co. (St. Louis, MO, USA). Other chemicals used in this experiment were analytical grade. The water was obtained from a Milli-Q^®^ (Millipore) water purification system (Billerica, MA, USA).

### 3.2. General

Optical rotations were made on a JASCO DIP-180 digital polarimeter. UV and IR spectra were recorded on a Shimadzu UV-1601PC and a Perkin-Elmer 983 G spectrophotometer, respectively. ^1^H- and ^13^C-NMR spectra were acquired on a Varian-Unity-Plus-400 spectrometer with residual solvent signals as internal reference. Chemical shifts are given in δ values and coupling constants (J) are given in hertz (Hz). EI-MS and HR-EI-MS were measured with a Jeol-JMS-HX300 mass spectrometer. Column chromatography (CC) was performed with silica gel (230–400 mesh; Merck & Co., Inc., Kenilworth, NJ, USA). TLC was performed with pre-coated silica gel plates (60 F-254; Merck & Co., Inc., Kenilworth, NJ, USA). Semi-preparative HPLC was performed using a normal phase column (Purospher STAR Si, 5 mm, 250 × 10 mm; Merck & Co., Inc., Kenilworth, NJ, USA) on a LDC Analytical-III system.

### 3.3. Plant Material

The bark of *C. japonica* D. Don was collected in Sitou, Taiwan in June, 2000. The plant material was identified by Dr. Yen-Hsueh Tseng, Department of Forestry, National Chung-Hsing University (NCHU). A voucher specimen (TCF13443) was deposited at the Herbarium of the Department of Forestry, NCHU, Taiwan.

### 3.4. Extraction and Isolation

The air-dried bark of *C. japonica* (16.0 kg) was extracted by maceration with MeOH (100 L) three times (seven days each time) at room temperature. After filtration, the combined MeOH extract was evaporated under reduced pressure to give a crude extract (480 g). The obtained extract was suspended in H_2_O (1 L), and successively partitioned with EtOAc (1 L) and *n*-BuOH (1 L) three times. The EtOAc soluble fraction (430 g) was loaded onto a silica gel (4.0 kg) column and eluted with *n*-hexane–EtOAc and EtOAc–MeOH mixtures to give 11 fractions, fr. 1 (2.6 g), 2 (29.4 g), 3 (47.8 g), 4 (92.4 g), 5 (21.6 g), 6 (18.1 g), 7 (22.5 g), 8 (35.8 g), 9 (19.2 g), 10 (44.2 g), and 11 (72.2 g). Fr. 3 from hexane/EtOAc (9:1) elution was further purified through a silica gel column (7 cm × 60 cm) and eluted with hexane/CH_2_Cl_2_ (1:0–0:1 *v*/*v*) to obtain nine fractions, 3A–3I. Further purification of subfraction 3E by HPLC gave 4 (15.1 mg) and 7 (1.5 mg) using hexane/EtOAc (9:1 *v*/*v*). Further purification of subfraction 3G by HPLC gave 3 (2.4 mg) using hexane/EtOAc (10:1 *v*/*v*). Further purification of subfraction 3H by HPLC gave 1 (1.6 mg) and 2 (3.3 mg) using hexane/EtOAc (9:1 *v*/*v*). Fr. 5 from *n*-hexane–EtOAc (7:3 *v*/*v*) elution was further purified over a silica gel column (5 cm × 45 cm), eluted with *n*-hexane–CH_2_Cl_2_–EtOAc (8:8:1 to 0:1:1 *v*/*v*/*v*) to yield fifteen fractions, 5A–5O. Further purification of subfraction 5E by HPLC gave 5 (0.5 mg) and 6 (1.9 mg) using *n*-hexane–EtOAc (7:3 *v*/*v*).

Abieta-6,8,11,13-tetraen-12-yl 12-hydroxyabieta-8,11,13-trien-7α-yl peroxide (1). Gum; [α${]}_{D}^{25}$: −21.2 (c = 0.7, CHCl_3_); EI-MS (70 eV) *m*/*z* (rel. int.): 584 ([M]^+^, 1), 300 (77), 284 (100), 269 (18), 227 (15), 213 (39), 202 (530), 189 (91); HR-EI-MS *m*/*z*: 584.4238 [M]^+^ (calculated for C_40_H_56_O_3_ 584.4232); UV (MeOH) λ_max_ (log ε): 220 (4.74), 279 (4.41) nm; IR (KBr) ν_max_: 3409, 3049, 1593, 1487, 1460, 1407, 1255, 1169, 1043, 1009, 890 cm^−l^; ^1^H-NMR and ^13^C-NMR (400/100 MHz, in CDCl_3_): See Table 1; chemical spectra is in the Appendix A.

Abieta-6,8,11,13-tetraen-12-yl 12-hydroxyabieta-8,11,13-trien-7β-yl peroxide (2). Gum; [α${]}_{D}^{25}$: −54.6 (c = 1.0, CHCl_3_); EI-MS (70 eV) *m*/*z* (rel. int.): 584 ([M]^+^, 1), 582 (3), 473 (7), 313 (5), 300 (45), 284 (100), 269 (9), 213 (15), 202 (30), 189 (25), 59 (14); HR-EI-MS: *m*/*z*: 584.4237 [M]^+^ (calculated for C_40_H_56_O_3_ 584.4232); UV (MeOH) λ_max_ (log ε): 217 (4.80), 276 (4.45) nm; IR (KBr) ν_max_ 3423, 3048, 1590, 1493, 1460, 1407, 1261, 1169, 1062, 1036, 1003, 963, 890, 738 cm^−1^; ^1^H-NMR and ^13^C-NMR (400/100 MHz, in CDCl_3_): See Table 1; chemical spectra is in the Appendix A.

12-Hydroxyabieta-8,11,13-trien-7β-yl 7-oxoabieta-5,8,11,13-tetraen-12-yl peroxide (3). Gum; [α${]}_{D}^{25}$: −39.7 (*c* = 0.8, CHCl_3_); EI-MS *m*/*z* 598 ([M]^+^, 1), 300 (50), 298 (26), 285 (63), 229 (47), 213 (41), 203 (37), 189 (100), 69 (37), 55 (44); HR-EI-MS *m*/*z* [M]^+^: 598.4018 [M]^+^ (calculated for C_40_H_54_O_4_ 598.4024); UV (MeOH) λ_max_ (log *ε*): 233 (4.34), 282 (4.08), 310 (3.88) nm; IR (KBr) ν_max_ 3376, 3049, 1639, 1586, 1467, 1407, 1275, 1182, 1036, 990, 897 cm^−1^; ^1^H-NMR and ^13^C-NMR (400/100 MHz, in CDCl_3_): See Table 1; chemical spectra is in the Appendix A.

### 3.5. Xanthine Oxidase Inhibition Assay

The inhibitory effect on xanthine oxidase of compounds 1–3 was measured spectrophotometrically according to the method reported by Chen et al. with minor modifications [43]. The mixture assay consisted of a 35 μL of 0.1 mM phosphate buffer (pH = 7.5), 30 μL of enzyme solution (0.01 units/mL in 0.1 mM phosphate buffer, pH = 7.5), and 20 μL of the sample solution (final concentration was 50 μM). The mixture was pre-incubated at 25 °C for 15 min, and then was initiated by adding 60 μL of substrate solution (150 mM xanthine in the same buffer). The reaction mixture was incubated for further 30 min at 25 °C. The reaction was stopped by adding 50 μL of 2 N HCl, and the absorbance was measured at 290 nm using a microplate reader. The percentage activity of xanthine oxidase was calculated as following the formula: XO Inhibition (%) = (1 − B/A) × 100, where A and B are the activities of the enzyme without and with test sample. Quercetin, a known inhibitor of xanthine oxidase, was used as a positive control, whereas a negative control was performed without any inhibitor.

### 3.6. Angiotensin-I Converting Enzyme (ACE) Inhibition Assay

ACE assay was performed using the modified spectrophotometric method described by Cushman and Cheung with minor modifications [44]. The assay mixture contained 30 µL 2.5 mM Hippuryl-l-histidyl-l-leucine (HHL), 10 µL of testing sample at a certain concentration, and 20 µL of ACE (0.05 mU/µL) in 200 mM borate buffer containing 300 mM NaCl (adjusted to pH 8.3). The mixture was incubated at 37 °C for 60 min, and then was halted by addition of 30 μL of 2 N HCl. The substrate HHL and product hippuric acid (HA) liberated through hydrolysis of HHL were determined by HPLC equipped with a Hypersil GOLD C-18 analytical column (250 mm × 4.6 mm, 5 μm). The column was eluted with a mobile phase of 23% ACN containing 0.1% TFA at a constant flow rate of 1 mL/min for 15 min and monitored for absorbance at 228 nm. The inhibition activity was calculated using the following formula: ACE Inhibition (%) = [1 − (∆A_Inhibitor_/∆A_Blank_)] × 100, where ∆A_Inhibitor_ and ∆A_Blank_ were the peak areas of HA in testing and blank samples, respectively. Captopril, a known inhibitor of ACE, was used as a positive control, whereas a negative control was performed without any inhibitor. 

## 4. Conclusions

In this study, three new dimeric abietanoid peroxides, abieta-6,8,11,13-tetraen-12-yl 12-hydroxyabieta-8,11,13-trien-7α-yl peroxide (**1**), abieta-6,8,11,13-tetraen-12-yl 12-hydroxyabieta-8,11,13-trien-7β-yl peroxide (**2**), and 12-hydroxyabieta-8,11,13-trien-7β-yl 7-oxoabieta-5,8,11,13-tetraen-12-yl peroxide (**3**), together with four known abietane-type diterpenoids (**4**–**7**) were isolated and characterized from the bark of *C. japonica*. At a concentration of 50 μM, the three new compounds exhibited the xanthine oxidase inhibitory activity. In addition, compounds **3** also showed ACE inhibitory activity.

## Figures and Tables

**Figure 1 molecules-24-02178-f001:**
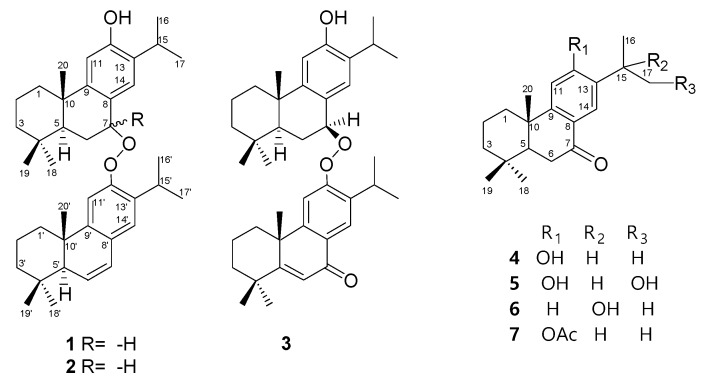
The chemical structures of compounds **1**–**7** isolated from *C. japonica*.

**Figure 2 molecules-24-02178-f002:**
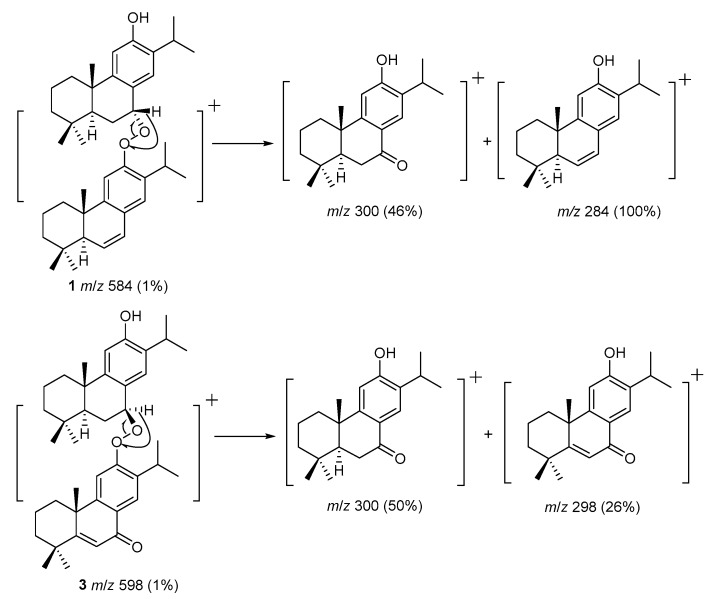
Some key electron impact (EI)-Mass fragmentations of compounds **1** and **3**.

**Figure 3 molecules-24-02178-f003:**
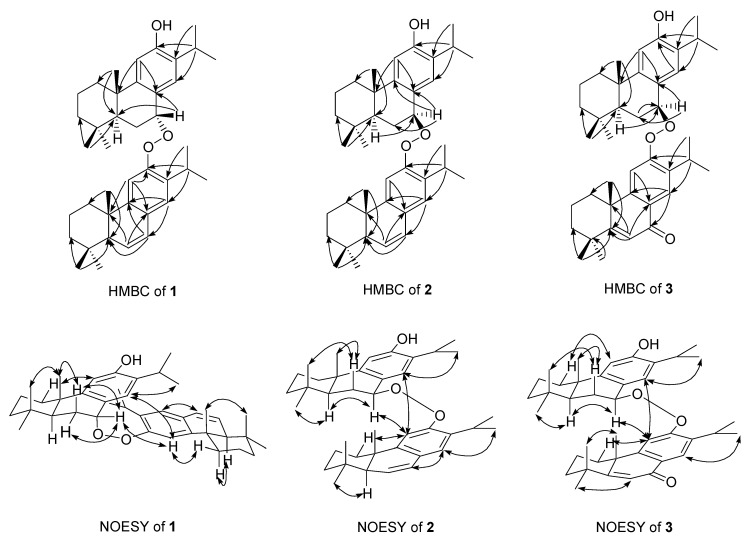
Significant HMBC (one-headed arrows) and NOESY (two-headed arrows) correlations of compounds **1**–**3**.

**Table 1 molecules-24-02178-t001:** NMR (nuclear magnetic resonance) data (CDCl_3_) of compound **1**–**3**. δ in ppm, *J* in Hz.

No.	1		2		3	
	δ_C_ ^a^	δ_H_ ^b^	δ_C_	δ_H_	δ_C_	δ_H_
1	41.4	1.78 m, 1.96 br d (12.0)	40.8	1.80 m, 1.93 br d (12.0)	41.0	1.78 m, 1.93 br d (12.1)
2	18.8	1.59 m	18.6	1.60 m	18.6	1.59 m, 1.77 m
3	42.8	1.30 td (13.0, 3.5), 1.50 m	42.6	1.25 m, 1.51m	42.4	1.25 m, 1.51 m
4	34.5		34.8		34.7	
5	43.5	1.96 m	47.6	1.59 m	47.3	1.72 m
6	34.0	2.28 m, 2.32 m	34.8	2.24 br d (14.0), 2.40 m	34.2	2.24 br d (14.3), 2.41 m
7	97.0	5.86 br s	104.1	5.20 d (8.0)	102.7	5.38 dd (9.0, 2.5)
8	145.6		147.2		146.0	
9	142.2		143.1		142.5	
10	40.6		40.7		41.5	
11	113.3	6.69 s	113.5	6.74 s	114.0	6.79 s
12	148.4		148.9		149.5	
13	131.9		132.8		133.1	
14	121.3	6.34 s	120.9	6.72 s	120.7	6.68 s
15	26.3	2.85 sept (7.0)	27.3	3.01 sept (7.0)	27.3	3.02 sept (7.0)
16	22.3	0.78 d (7.0)	22.6	1.18 d (7.0)	22.5	1.16 d (7.0)
17	22.2	1.01 d (7.0)	21.7	1.16 d (7.0)	21.7	1.11 d (7.0)
18	33.5	0.97 s	33.8	0.95 s	33.7	0.97 s
19	23.2	0.98 s	23.1	1.02 s	23.1	1.01 s
20	21.9	1.36 s	21.4	1.37 s	21.7	1.36 s
12-OH		4.33 s		4.52 s		4.52
1′	36.2	1.55 m, 2.22 br d (13.0)	36.3	1.65 m, 2.13 m	38.0	1.72 m, 2.51 br d (13.5)
2′	19.0	1.75 m, 1.65 m	18.8	1.69 m	18.7	2.00 br d (13.5), 1.70 m
3′	41.0	1.23 m, 1.51m	40.9	1.23 m, 1.51m	40.3	1.45 m, 1.70 m
4′	32.8		32.8		37.5	
5′	51.1	2.07 dd (3.0, 2.5)	51.1	2.13 dd (3.0, 2.0)	173.2	
6′	127.6	5.87 dd (9.5, 2.5)	128.0	5.90 dd (9.5,3.0)	124.5	6.48 s
7′	127.3	6.45 dd (9.5, 3.0)	127.4	6.51 dd (9.5,2.0)	185.4	
8′	126.5		127.3		124.5	
9′	146.6		146.9		153.5	
10′	38.1		37.9		41.4	
11′	107.7	6.98 s	109.1	6.95 s	109.6	7.19 s
12′	152.9		153.6		158.2	
13′	133.6		135.4		136.9	
14′	124.4	6.77 s	124.5	6.92 s	124.5	8.03 s
15′	27.0	2.82 sept (7.0)	26.5	3.34 sept (7.0)	26.9	3.32 sept (7.0)
16′	21.9	0.81 d (7.0)	22.9	1.21 d (7.0)	22.6	1.23 d (7.0)
17′	22.2	0.98 d (7.0)	22.8	1.19 d (7.0)	22.5	1.20 d (7.0)
18′	32.6	0.94 s	32.6	0.97 s	32.6	0.97 s
19′	22.5	1.04 s	22.5	1.03 s	29.0	1.03 s
20′	20.5	1.11 s	23.0	1.00 s	32.3	1.49 s

Recorded at ^a^ 100 MHz (^13^C); and ^b^ 400 MHz (^1^H).

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
