# Peer review of "Bioactive Dimeric Abietanoid Peroxides from the Bark of Cryptomeria japonica"

_molecules, 2019, doi:10.3390/molecules24112178_

Round 1

Reviewer 1 Report

see comments below

this is a nice piece of research in Natural products, compounds have been properly isolated and characterized using state of the art NMR techniques.

I would only suggest the authors to improve the biological activity discussion (section 2.2), since the discussion is really reduced to the minimum: actually, less than three lines if you analyze adequately the content of that section

I have the perception that some English improvements are required but as a non-native speaker I leave this to the editorial board.

Author Response

Thank you for your suggestions. On pages 6-7, we describe more about the results of the biological activities of the manuscript.

Reviewer 2 Report

This manuscript reported that three new dimeric abietane-type diterpenoids from the methanol extract of the bark of Cryptomeria japonica, and their enzyme inhibitory activities. The research design is appropriate and the results are clearly presented. Some supplement should be required before the acceptance of the manuscript:

In the SUPPLEMENTARY MATERIAL, please provide the HPLC separation chromatogram of compounds 1 and 2.
Please compare and explain the NMR data differences between compounds 1 and 2, for example C-5 and C-20’.
Please supplement the Circular Dichroism (CD) of compounds 1 and 2, and explain whether the absolute configuration can be determined.
Please supplement the 1H and 13C NMR spectra of compound 4-7 in the SUPPLEMENTARY MATERIAL.

Page 7, EtOAc or AcOEt should be uniform.

Author Response

1.     We have added the HPLC separation chromatogram of compounds 1 and 2 into the SUPPLEMENTARY MATERIAL.

2.     We have made more description on the NMR data differences between compounds 1 and 2 on page 4.

3.     Thank you for your suggestions. Since we have no reference compounds, (R)-7-hydroxyferruginol and (S)-7-hydroxyferruginol, for the determination of absolute configuration of compounds 1 and 2 using Circular Dichroism, we can only confirm the relative configuration of compounds 1 and 2 according to their NOESY spectral data.

4.     We have added the 1H NMR spectra of compounds 4-7 into the SUPPLEMENTARY MATERIAL.

5.     We have revised the manuscript.

Round 2

Reviewer 1 Report

the authors have addressed my requirements